# Cadmium (Cd) and Nickel (Ni) Distribution on Size-Fractioned Soil Humic Substance (SHS)

**DOI:** 10.3390/ijerph16183398

**Published:** 2019-09-13

**Authors:** Sheng-Hsien Hsieh, Teng-Pao Chiu, Wei-Shiang Huang, Ting-Chien Chen, Yi-Lung Yeh

**Affiliations:** 1Department of Civil Engineering, National Pingtung University of Science and Technology, Pingtung 91201, Taiwan; tino388@yahoo.com.tw (S.-H.H.); ctp5559@gmail.com (T.-P.C.); 2Department of Environmental Science and Engineering, National Pingtung University of Science and Technology, Pingtung 91201, Taiwan; stefsun921015@yahoo.com.tw (W.-S.H.); chen5637@mail.npust.edu.tw (T.-C.C.)

**Keywords:** soil humic substances (SHS), cadmium and nickel, SHS separation, [Me]/[DOC] ratios, optical indicators, multilinear regression

## Abstract

Soil humic substances (SHS) are heterogeneous, complex mixtures, whose concentration, chemical composition, and structure affect the transport and distribution of heavy metals. This study investigated the distribution behavior of two heavy metals [cadmium (Cd) and nickel (Ni)] in high molecular weight SHS (HMHS, 1 kDa–0.45 μm) and low molecular weight SHS (LMHS, <1 kDa) extracted from agricultural soils. The HMHS mass fractions were 45.1 ± 19.3%, 17.1 ± 6.7%, and 57.7 ± 18.5% for dissolved organic carbon (DOC), Cd, and Ni, respectively. The metal binding affinity, unit organic carbon binding with heavy metal ratios ([Me]/[DOC]), were between 0.41 ± 0.09 μmol/g-C and 7.29 ± 2.27 μmol/g-C. Cd preferred binding with LMHS (*p* < 0.001), while Ni preferred binding with HMHS (*p* < 0.001). The optical indicators SUVA_254_, S_R_, and FI were 3.16 ± 1.62 L/mg-C/m, 0.54 ± 0.18 and 1.57 ± 0.15, respectively for HMHS and 2.65 ± 1.25 L/mg-C/m, 0.40 ± 0.17, and 1.68 ± 0.12, respectively for LMHS. The HMHS contained more aromatic and lower FI values than LMHS. Multilinear regression showed a significant positive correlation between the measured predicted [Me]/[DOC] ratios (r = 0.52–0.72, *p* < 0.001). The results show that the optical indices can distinguish the chemical composition and structure of different size SHS and predict the binding ability of Me-SHS.

## 1. Introduction

Soil organic matter (SOM) is an important component of soil [1,2,3,4]. Heavy metals in soil are mainly associated with SOM and fine particles, such as oxide-type components and clay minerals [1,5,6,7]. The SOM’s major components are soil humic substances (SHS). The SHS is a heterogeneous and complex organic substance comprised of various functional groups such as phenol, carboxyl, and hydroxyl functional groups [1,5,7,8,9,10], which have a strong ability to bind heavy metals. When heavy metals bind with SHS, the complex affects the heavy metal mobility, biotoxicity, and fate in the soil environment [1,4,5,6,7,9,10].

The SHS concentration (represented by C content); however, does not fully predict the mobility potential and binding strength of metals. The SHS chemical composition, structure, and molecular weight are important factors that affect the Me-SHS binding capacity. In strongly-binding Me-SHS complexes, metals are stable and will not readily release; this reduces the metal mobile potential and biological toxicity. However, in a weakly-binding Me-SHS complex, metals may release into the environment and affect ecological safety and security, and inhibit crop growth [1,7,9,10,11,12]. Several studies have used the metal binding affinity [Me]/[DOC] ratio to understand the preferences, distribution, bioavailability, and mobile potential of metal binding with dissolved organic matters (DOM) and SHS [1,9,10,13,14,15].

The SHS has a wide range of molecular weights [1,2]. The SHS composition and molecular weight are greatly influenced by the source of the organic matter and the biogeochemical processes in the soil [1,2,11,16]. The SHS molecular weight may play an important role in heavy metal binding. A molecular weight of 1 kDa is a common quantity used to distinguish high and low molecular weight of DOM and SHS. Hence, in this study, SHS molecular weight less than 1 kDa was designated as the low molecular weight SHS (LMHS), and the molecular weight higher than 1 kDa was designated as the high molecular weight SHS (HMHS). It is worth noting that the LMHS still contained various heavy metal species (free ion metals, binding with low molecular weight SHS) [17,18,19,20,21,22,23]. Moreover, the low molecular weight DOM and SHS had a metal and hydrophobic organic compound binding capacity [24,25,26].

Previous studies on the distribution and partition of metals between high and low molecular weight organic matter focused on the DOM in an aquatic environment [17,21,22,23,27,28,29] and a few studies focused on soil solutions [18]. The optical indices were used to examine the influencing factors of the [Me]/[DOM] ratio, including a water/calcium chloride (CaCl_2_)-extracted soil solution and soil leachate [9,10] as well as bulk water DOM [13,14,15]. The widely-ranging SHS molecular weights had different chemical compositions and structure, which may have affected the Me-SHS binding behavior. Information on the distribution and partition of heavy metals in HMHS and LMHS separated by SHS is still lacking.

Optical methods (UV/Vis and fluorescence spectroscopy) are rapid, sensitive, non-destructive methods used to investigate DOM and SHS chemical composition and structure [30,31,32,33,34,35,36]. This study investigated (1) heavy metals (Cd and Ni) distribution in alkaline-extracted soil humic substances (BHS), and both size-fractioned HMHS (1 kDa–0.45 μm) and LMHS (<1 kDa), (2) UV-Vis and fluorescence indices (SUVA_254_, S_R_, and FI) used to investigate BHS, HMHS and LMHS chemical composition and structure, and (3) the correlation method examining the [Me]/[DOC] ratio-affecting factors in term of the SHS optical indicators.

## 2. Research Method and Material

### 2.1. Site Description and Soil Sampling, Treatment, and Measurement

The study site was located in a paddy experimental field in southern Taiwan (location 23°12′39.5″ N, 120°10′54.0″ E). The soil samples were taken with an auger (ID 5 cm) to 60 cm depth and about 2.0 kg was taken for each sample. Eight soil samples were collected from the sampling site each time. In total, 24 soil samples were used to analyze the concentration and optical values of the SHS and heavy metals.

The soil samples were air-dried (about one week) in the laboratory. The soil was passed through a sieve mesh #10 (2.0 mm) and provided for subsequent tests. The basic properties of the soil were measured with the sieved soil (pH, and organic matter (OM)). The soil contained 46%, 33%, and 21% sand, silt, and clay, respectively and was classified as loam soil. The soil pH was 8.07 ± 0.38 and the OM was 2.28 ± 0.09%. The total metal concentrations (Fe, Mn, Cd, Cr, Cu, Ni, and Zn) were analyzed with an aqua regia digestion method. Three g soil was added in 30 mL aqua regia solution; after 16 h settling and 2 h digestion at 180 °C, the solution was measured as 100 mL. Alkaline metals (Ca, Mg, K, and Na) were extracted with a 1.0 M ammonium acetate solution at soil mass/volume = 1/10.

### 2.2. HS Extraction and Separation

Bulk SHS (BHS) was extracted with an alkaline solution. Briefly, 5 g soil was added to 0.1 N HCl 100 mL solution to remove alkaline metals and carbonate. The 100 mL of 0.1 N NaOH was added to the residual soil at the W/V = 1/20. After 24 h shaking, the soil solution was centrifuged at 4500 rpm for 30 min. The NaOH-extracted humic substance was identified as the bulk SHS (BHS) solution. A three-liter BHS solution was used to separate the HMHS and LMHS fractions. A cross-flow ultrafiltration system equipped with 1 kDa nominal molecular weight cutoff ceramic membrane cartridges (Filtanium, France, membrane area 320 cm^2^) was used to separate the high molecular weight SHS (HMHS, 1 kDa–0.45 μm) and the low molecular weight SHS (LMHS, <1 kDa). The feed flow rate was 1.7–2.0 L/min and the penetration-flow rate was 25 mL/min. The cross-flow ultrafiltration working pressure was kept at 5 kg/cm^2^. In the concentration process, the volume concentration factor (C_f_) was kept at 10 (Equation (1)), in which the retentate flow was directed back to the feed bottle; this was the HMHS solution at the end of the separation. The penetration flow was directed to another container to be collected as the LMHS (<1 kDa). The mass balances of the dissolved organic carbon (C content) and heavy metals were calculated by Equation (2). The HMHS and LMHS mass fractions were calculated following Equations (3) and (4).
(1)$$C_{f}=\frac{V_{\mathrm{LMHS}}+V_{\mathrm{HMHS}}}{V_{\mathrm{HMHS}}},$$
(2)$$R\left(\%\right)=\frac{C_{\mathrm{HMHS}}\times V_{\mathrm{HMHS}}+C_{\mathrm{LMHS}}\times V_{\mathrm{LMHS}}}{C_{\mathrm{bulk}}\times V_{\mathrm{bulk}}}\times 100,$$
(3)$$M_{\mathrm{HMHS}}\left(\%\right)=\frac{C_{\mathrm{HMHS}}\times V_{\mathrm{HMHS}}}{C_{\mathrm{HMHS}}\times V_{\mathrm{HMHS}}+C_{\mathrm{LMHS}}\times V_{\mathrm{LMHS}}}\times 100$$
(4)$$M_{\mathrm{LMHS}}\left(\%\right)=100-M_{\mathrm{HMHS}}\left(\%\right)$$
where C_f_ was the volume concentration factor. R(%) was the mass balance percentage of the separated mass (HMHS added to LMHS) divided by the feed mass (BHS) for DOC, Cd, and Ni. C_HMHS_, C_LMHS_, and C_bulk_ were the measured metal concentrations and C content in the HMHS, LMHS, and BHS solutions, respectively. V_HMHS_, V_LMHS_, and V_bulk_ were the metal and DOC volumes for the separation procedure for the HMHS, LMHS and BHS solutions, respectively. M_HMHS_ and M_LMHS_ were the mass fractions of DOC and metals in the HMHS and LMHS solutions. The membrane was cleaned and preconditioned before each SHS separation experiment as recommended by manufacture protocol. The three SHS fractions were measured via DOC, UV-vis, and fluorescence spectroscopy.

### 2.3. Dissolved Organic Carbon and Metals Measurement

The BHS, HMHS, and LMHS C contents were measured with a TOC-V analyzer (Shimadzu, Japan). High concentrations of heavy metals and alkaline metals (Ca, Mg, K, and Na) were measured with an atomic absorption spectroscopy (AAS) (Hitachi, Z-2300, Japan). When the metal concentration was lower than the AAS detection limit, the metal concentration was measured with graphite furnace atomization (Hitachi, Z-3000, Japan).

### 2.4. UV-Vis and Fluorescent Measurement

The BHS and sized SHS solutions were diluted to 5 mg-C/L with ultrapure water. The absorbance was measured with an ultraviolet/visible spectrophotometer (Hitachi, U-2900) and fluorescence spectra were recorded on a fluorescence spectrometer (Hitachi F-7000). The absorbance at 700–800 nm was set as the background value. The absorbance of the sample was subtracted from the average of the absorbance at 700–800 nm [30]. The UV-Vis spectrophotometric scanning wavelength was 800–200 nm.

The excitation/emission matrixes (EEMs) were generated by recording emission spectra from 250–550 nm at 2.0 nm steps for an excitation wavelength between 200–450 nm at 5 nm increments. The scanning rate was 2400 nm/min. The value of the blank sample was subtracted from the sample fluorescent data. The UV-Vis absorbance at 254 nm was <0.2 and the inner filter effect correction was ignored.

### 2.5. Optical Index and Metal Binding Affinity Calculation

The specific ultraviolet absorbance at 254 nm (SUVA_254_, L/mg-C/m) is the absorbance of the sample at 254 nm UV_254_ (cm^−1^) divided by the C content of the SHS sample (mg-C/L) multiplied by 100 [36]. The UV-Vis spectral slopes in the 275–295 nm and 350–400 nm spectral ranges were determined and were reported as S_275–295_ to S_350–400_. The slope ratio S_R_ was calculated as the ratio of S_275–295_ to S_350–400_. [30,37]. The Fluorescence Index (FI) is the fluorescence intensity ratio of Em = 450 to Em = 500 nm at Ex = 370 nm [31,32]. The metal binding affinity ([Me]/[DOC] ratio, μmol-Me/g-C) was calculated as the metal concentration divided by the SHS C content.

### 2.6. Statistic Analysis

In this study, linear regression, step-wise regression, and the difference tests used the S-Plus software (V 6.2, Insightful Corporation, Seattle, WA, USA) at significance levels at *p* < 0.05. Two group difference tests between HMHS and LMHS (such as, concentration, mass fraction, [Me]/[DOC] ratio, and indicators) were used in the t-test method. The three group difference tests used the ANOVA test method. Fluorescence indicators were calculated at R script developed by Lapworth and Kinniburgh [38]. The step-wise procedure, the measured [Me]/[DOC] ratio was set as the dependent parameter and the selected indicators were set as independent parameters. We then conducted the step-wise regression procedure to obtain the significant indicator and coefficients. Next, we used the predicted equation and significant indicators and coefficients to calculate the predicted [Me]/[DOC] ratio.

## 3. Results and Discussion 

### 3.1. UV-Vis and Fluorescent Index

Optical indicators have been used extensively and are effective tools for examining the chemical composition and structure of DOM and dissolved SHS [3,9,10,16,39,40]. Table 1 lists the SUVA_254_, S_R_, and FI values of BHS, HMHS, and LMHS. The SHS optical indices were comparable to reported values in soil solution, where SUVA_254_ values ranged from 0.38–7.18 L/mg-C/m [3,9,10,16,40], S_R_ values ranged from 0.64–3.26 [16,39], and FI values ranged from 1.08–2.03 [3,16,31].

The SUVA_254_ is positively correlated with the aromatic content [41]. When SUVA_254_ is <3 L/mg-C/m, the SHS solution consists mainly of hydrophilic aromatic compounds, while when SUVA_254_ is >4 L/mg-C/m, the composition of the SHS solution is dominated by hydrophobic compounds [34]. The average HMHS SUVA_254_ value was higher than the average LMHS SUVA_254_ value but insignificantly different (*p* = 0.23). The SUVA_254_ value showed that the SHS solution mainly contained hydrophilic and low aromatic compounds.

S_R_ is the indicator of average molecular weight [30]. S_R_ values show that HMHS was significantly greater than LMHS (*p* = 0.007). Although, Helms et al. [30] suggested that S_R_ was negatively correlated with mean molecular weight, the S_R_ of this study was significantly positively correlated with the SUVA_254_ and significantly negatively correlated with the FI, respectively (r = 0.842 for SUVA_254_, r = −0.765 for FI, *p* < 0.001, *n* = 72). HMHS had higher S_R_ values than LMHS (Table 1), which confirmed that higher S_R_ values had higher aromaticity, molecular weight and terrestrial sources in this study.

FI is the relative contribution of terrestrial sources [3,31,32,42]. FI values less than 1.4 indicated the HS had strong terrestrial sources and when larger than 1.9, had microbial sources [31,32]. HMHS and LMHS both had median terrestrial sources and HMHS had a higher terrestrial contribution than LMHS (*p* = 0.006). FI values had a significantly negative correlation with SUVA_254_ values (r = −0.735, *p* < 0.001). The optical indices showed that the HMHS had higher aromaticity, average molecular weight, and terrestrial sources than LMHS.

### 3.2. Soil Properties and Metal Concentration

The soil basic properties of pH, OM, and TOC were 8.28 ± 0.08%, 3.12 ± 0.32%, and 1.81 ± 0.19%, respectively for the 24 samples. The total Ca and Fe concentrations were 1.44 ± 0.18 g/kg and 21.8 ± 4.5 g/kg, respectively. The concentrations of Mg, K, and Na were 316 ± 30, 395 ± 47, 64.7 ± 12.7, and 55.3 ± 5.7 mg/kg, respectively. The basic soil properties and main metal total concentrations were similar to those of general farm land soil [4,12,43,44].

The BHS C content was 678 ± 247 mg/kg and the BHS-C content/TOC ratio was 3.7%. In four agricultural soil tests by Fernández-Romero et al. [3], organic matter ranged from 4.3–15.6%. The water extraction organic carbon (WEOC) was 80–620 mg/kg at room temperature; the WEOC/OM ratios were 0.12–0.40%. In another study, Nkhili et al. [40] reported soil was extracted with different water temperature (20, 60, and 80 °C). The WEOC was 310–401 mg/kg, and WEOC/TOC ratios were 0.93–1.16%. The alkaline-extracted soil organic carbon in the study was slightly higher than water-extracted soil organic carbon at room temperature.

Table 2 lists the total concentrations of Cd, Cr, Cu, Ni, and Zn and the BHS concentrations of these heavy metals. The total metal concentrations are generally similar to the concentration of the uncontaminated agricultural soils [4,11,43,44]. The BHS concentration of Cu (1.19 mg/kg) was higher than the concentrations of the other metals; all other metals were below 1.0 mg/kg. The ratios of BHS to the total metal concentration shows that the ratios of Cd to Cu had higher ratios (15.4% and 12.1%, respectively) than the other metals Ni (1.07%), and both Cr and Zn < 1%. Cambier et al. [11] reported that “the ratios suggested that Cu and Cd preferentially combined with soil humic substances.” Matong et al. [4] reported that three agricultural soils were sequentially extracted with acetic, ascorbic and hydrogen peroxide digestion. The exchangeable, and organic binding fractions were 52–61%, and 24–32% for Cd, and Ni, respectively. The fractions were similar to our results that showed Cd was higher than Ni by exchangeable and organic fraction.

### 3.3. Metal and SHS C Content and Distribution between HMHS and LMHS

Table 3 lists the C content and concentrations of Cd and Ni of BHS, HMHS, and LMHS. This study investigated the metals Cd and Ni because they had accepted metal mass balance. However, the number of mass balances within reasonable range (100 ± 25%) [22,26,27,45] were 15, 10, and 6 for Cr, Cu, and Zn, respectively for the separation. We ignored these data and the distribution of heavy metals Cr, Cu and Zn was not investigated in later discussions. The mass balances of DOC, Cd, and Ni in the separation process were calculated following Equation (2). The average mass balances were between 83–90%, which were within a reasonable range.

Wu and Tanous [45] separated lake water into three size-fractioned DOM, which included 0.1–0.7 μm, 5 kDa–0.1 μm, and <5 kDa fractions. The carbon mass balances were 89–109%. Knoth de Zarruk et al. [46] reported that the four water-extracted soil and waste-borne DOM were separated into four size-fractioned DOM where the DOC mass balances were 80–151%. Martin et al. [17] reported the DOC mass balances ranged from 85–98% separated with lagoon water DOM. Luan and Vadas [28] separated storm water and effluent DOM and the mass balances were 101–105% for Cd [28]. Dabrin et al. [22] reported that five-sediment pore water DOM was separated into high molecular weight (5 kDa–0.45 μm) and low molecular weight DOM (<5 kDa). The Cd and Ni mass balances were 50–220% (one exception 2000%), and 100–280%, respectively.

The average mass fractions of DOC, Cd, and Ni for HMHS and LMHS (Table 3, calculated by Equations (3) and (4)). The LMHS mass fractions were 82.9 ± 6.7% and 55.9 ± 19.3% for Cd and DOC, respectively. They were significantly greater than the HMHS mass fractions (17.1 ± 6.7% (Cd), *p* < 0.001; 44.1 ± 19.3% (DOC), *p* = 0.038). However, the Ni HMHS mass fraction (57.7 ± 18.5%) was significantly higher than the LMHS mass fraction (42.3 ± 18.5%, *p* = 0.006). The mass fraction shows that Cd preferred to bind with LMHS, and Ni preferred to bind with HMHS.

Wang et al. [18] used a mixture of acetic acid (formic, and malic acid) to extract soil solutions, which were separated into HMHS (1 kDa–0.45 μm) and LMHS (<1 kDa). Most metals had a high LMHS fraction (71% (Cd) and 82% (Ni)). Ilina et al. [47] used ultrafiltration to separate water-extracted soil solution; the HMHS were 65%, and 83% for Cd, and Ni, respectively. Hartland et al. [48] observed hyperalkaline cave drip-water, the HMW fractions (1 kDa–1.0 μm) were 64% for Ni, which were similar to the results in our study. In four treatment processes in a municipal wastewater treatment plant, Hargreaves et al. [23] reported the HMW fractions (1 kDa–0.45 μm) were 67–75% for Ni [23]. In sediment pore water, Dabrin et al. [22] reported the HMW fractions (5 kDa–0.45 μm) were 9–18%, and 5–20% for Cd, and Ni, respectively (estimated from Figure). The mass fractions of high and low molecular weights in water extracted organic matter and DOM are variations. The fractions may depend on the matrix, type of metal, extraction solvent, extraction method and solid/liquid ratio, and separation method and conditions [18,49].

### 3.4. [Me]/[DOC] Ratio

The metal binding affinity, [Me]/[DOC] ratio, was used to understand the difference in the binding ability of metals to DOM and WEOM [9,10,13,14,15]. Table 3 shows the [Me]/[DOC] ratios of Cd and Ni in BHS, HMHS, and LMHS solutions. The [Me]/[DOC] ratios of HMHS to LMHS show that the ([Ni]/[DOC])_HMHS_ ratio was significantly greater than ([Ni]/[DOC])_LMHS_ (*p* < 0.001), but the ([Cd]/[DOC])_HMHS_ ratio was significantly less than the ([Cd]/[DOC])_LMHS_ (*p* < 0.001). The [Me]/[DOC]_BHS_ ratio was higher than the soil WEOM [Me]/[DOC] ratio as reported by Sauvé et al. [8]. The [Me]/[DOC] ratios were 0.025, and 015 μmol/g-OC ratios for Cd and Ni, respectively in a forest soil WEOM solution. The low ratios were due to high soil carbon content (50.1%) in the soil. The [Ni]/[DOC] and [Cd]/[DOC] ratios were comparable to natural water DOM samples reported by Baken et al. [13], which were 1.6 ± 0.0 to 4.1 ± 1.5 μmol/g-OC for Ni, and 1.2 ± 0.2 to 3.0 ± 0.2 μmol/g-OC for Cd; however, the [Me]/[DOC] ratios were lower than those impacted by anthropogenically-influenced DOM samples, which were 5.8 ± 1.3 to 14.7 ± 0.2 μmol/g-OC for Ni and 4.4 ± 0.6 to 12.5 ± 0.3 μmol/g-OC for Cd. In a WWTP final effluent, Hargreaves et al. [23] reported that the [Ni]/[DOC] ratios were 4.6 and 8.3 μmol/g-OC for the low (<1 kDa) and high (>1 kDa) molecular weight DOM.

### 3.5. Correlation between [Me]/[DOC] Ratios and Optical Indices

We used the Pearson correlation analysis to explore the SHS factors affecting the [Me]/[DOC] ratios. Table 4 shows the correlation coefficients between [Me]/[DOC] ratios and selected indicators for BHS, HMHS, and LMHS. The ratios of [Cd, Ni]/[DOC]_BHS_ and [Cd, Ni]/[DOC]_LMHS_ had a significant correlation with the C content, indices S_R_ and SUVA_254_ (*p* < 0.02). Furthermore, [Cd]/[DOC]_LMHS_ had a significantly negative correlation with the index FI (*p* = 0.047). However, there was no significant correlation between [Me]/[DOC]_HMHS_ and optical indicators or the C content.

The ratios of [Me]/[DOC]_HMHS_ had no correlation with the HMHS optical indicators, which may be due to the complex composition of HMHS containing biopolymers [42]. It is believed that humic acid (HA) and fulvic acid (FA) contain various functional groups that have a strong metal-binding ability. The HA and FA molecular weights range from sub-kDa to several kDa [1,42]. The LMHS indices S_R_ and SUVA_254_ had a strong correlation with [Cd, Ni]/[DOC] ratios, which suggested that LMHS (<1 kDa), still containing low molecular weight HA and FA, have the potential to bind to Cd and Ni. Meanwhile, the ([Cd]/[DOC])_LMHS_ ratio had a negative correlation with the terrestrial sources index, which suggested that the HS high terrestrial sources had a high Cd-HS binding ability [50,51]. The ratios of [Cd, Ni]/[DOC]_HMHS_ were insignificantly correlated with the indices, which implied a complex composition of HMHS, with aromatic content and molecular weight, cannot predict the Me-HS binding ability. The high molecular weight sources may contain biopolymers.

Table 4 lists the correlation coefficients between the combined optical indices of BHS, HMHS, and LMHS and the [Me]/[DOC] ratios of the three sized solution. The results showed that the [Cd]/[DOC] ratio was only weakly positive when correlated with SUVA_254_ and was insignificantly correlated with the FI and S_R_ indicators. The [Ni]/[DOC] ratios have a weak to moderate correlation with the three selected indicators. High aromaticity, large molecular weight, and a high terrestrial source contribution in SHS increased the Ni-SHS binding ability. The [Cd, Ni]/[DOC] ratios were significantly negative when correlated with C content as previously reported [9,10].

Previous studies reported that the correlation between the [Me]/[DOC] ratio and the optical indicators mainly focused on the aqueous phase and soil bulk solution. For example, the [Cu]/[DOC] ratio in soil water-extracted organic matter had a significant positive correlation with SUVA_254_ and a negative correlation with C content [9,10]. In natural water DOM, ratios of [Cu, Fe, Zn]/[DOC] had a significantly positive correlation with SUVA_254_ values. However, in water DOM affected by effluent of WTP, the [Me]/[DOC] ratios did not show correlations with SUVA_254_ [15]. In studied natural water DOM, influenced by anthropogenic input, the DOM tests demonstrated that the SUVA_254_ values of natural water DOM had a significant positive correlation with ratios of [Cd, Cu, Ni, Zn]/[DOC]; however, anthropogenic-affected DOM SUVA_254_ values had no correlation with ratios of [Cd, Cu, Ni, Zn]/[DOC] [13].

We used a step-wise multilinear regression to investigate the correlations between the total indicators and total [Me]/[DOC] ratios in combined BHS, HMHS, and LMHS solutions. The multilinear regression equation is shown in Table 5. The predicted [Me]/[DOC] ratios used the regression results with optical indices; the measured [Me]/[DOC] ratios are shown in Figure 1a,b. The correlation coefficients between the predicted [Me]/[DOC] ratio used multiple indicators from multilinear regression. Table 5 shows two types of regression equations; one was predicted with the three selected indicators and the other equation predicted with the three indicators and the C content. When the C content was added as a predicted indicator, the determined coefficients were slightly increased compared to the regression without the C content. The C content coefficient was negative, which is the same as the result in Table 4 and is consistent with the previous study [9,10].

The [Cd]/[DOC] ratio predictive equation shows that the SUVA_254_ coefficient had a positive value and the S_R_ had negative value. However, the [Ni]/[DOC] ratio predictive equation coefficients were in contrast to the [Cd]/[DOC] ratio predictive equation; the SUVA_254_ coefficients had a negative value and S_R_ had a positive value. The reverse coefficients for Cd and Ni suggested that the influencing Me-HS binding factors could be different for different metals. In the studied HS, Cd preferred combining with low molecular weight HS, so it is negatively correlated with the molecular weight indicator (S_R_) and positively correlated with aromaticity (SUVA_254_). However, Ni preferred combining with high molecular weight; therefore, it positively correlated with the molecular weight indicator (S_R_) and negatively correlated with aromaticity (SUVA_254_). The coefficient values of FI for both Cd and Ni were positive, which indicated that higher terrestrial sources correlated with lower Me-HS binding ability. The Cd and Ni preferred to combine with the SHS microbial source. The results of the [Me]/[DOC] ratio predicted by the indicators show that the combination of Me-SHS is affected by both the C content and the chemical structure of the SHS.

## 4. Conclusions

Separating soil SHS into different molecular weights allows understanding of the different chemical structure and composition of HS and HS-Me binding ability between the HMHS and LMHS. The Cd and Ni were not evenly distributed in HMHS and LMHS. The separated SHS showed that Cd favored binding with LMHS, but Ni favored binding with HMHS. The optical indices had a significant correlation with the [Me]/[DOC] ratios of BHS and LMHS. However, the [Me]/[DOC] ratio of HMHS had no significant correlation with the optical indices. Possibly, because of the complex composition of HMHS, aromaticity and a terrestrial source cannot explain the Me-SHS binding ability. The multilinear regression used with the selected optical indices were able to predict the measured [Me]/[DOC] ratios. In addition, the indicator coefficient affected the [Cd, Ni]/[DOC] ratios and had a reverse relationship. The results show that the optical indices can distinguish the chemical composition and structure of different size SHS and predict the binding ability of Me-SHS.

## Figures and Tables

**Figure 1 ijerph-16-03398-f001:**
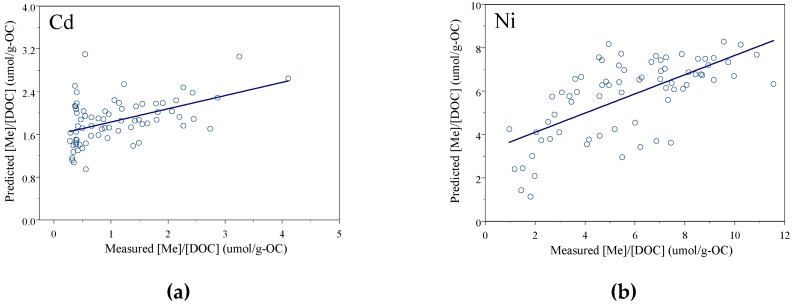
The linear relationship of the measured [Me]/[DOC] ratio and predicted [Me]/[DOC] ratio using the selected indicators for Cd (**a**) and Ni (**b**).

**Table 1 ijerph-16-03398-t001:** Optical indicators of three sized soil humic substances solutions (*n* = 24).

Samples	SUVA_254_ (L/mg-C/m)	S_R_ *	FI *
**BHS**	3.06 ± 1.32	0.52 ± 0.16	1.63 ± 0.13
**HMHS**	3.16 ± 1.63	0.54 ± 0.18	1.57 ± 0.15
**LMHS**	2.65 ± 1.25	0.40 ± 0.17	1.68 ± 0.12

* *p* < 0.05; between high and low molecular weight soil humic substances

**Table 2 ijerph-16-03398-t002:** Total and extracted bulk soil humic substances BHS heavy metal concentrations (mg/kg, *n* = 24).

	Cu	Cd	Cr	Ni	Zn
**Total**	9.89 ± 0.69	0.473 ± 0.022	19.3 ± 2.4	19.6 ± 1.8	73.2 ± 28.5
**BHS**	1.19 ± 0.40	0.073 ± 0.026	0.06 ± 0.04	0.21 ± 0.08	0.36 ± 0.50

**Table 3 ijerph-16-03398-t003:** Cadmium (Cd) and Nickel (Ni) and C content in BHS, HMHS, and LMHS solutions (*n* = 24).

Samples	DOC (mg/L)	Cd (μg/L)	Ni (μg/L)
**BHS**	(33.9 ± 12.4) ^a^	(3.65 ± 1.29) ^a^, 1.04 ± 0.38 ^b^	(10.4 ± 3.8) ^a^; 5.81 ± 2.38 ^b^
**HMHS**	(12.2 ± 6.1, 44%) ^a^	(0.55 ± 0.25, 17%) ^a^; 0.41 ± 0.09 ^b^	(5.1 ± 2.6, 58%) ^a^; 7.29 ± 2.27 ^b^
**LMHS**	(17.2 ± 10.3, 56%) ^a^	(2.61 ± 0.74, 83%) ^a^; 1.82 ± 0.95 ^b^	(3.6 ± 2.1, 42%) ^a^; 4.22 ± 2.38 ^b^
**Mass balance**	90 ± 26%	89 ± 14%	83 ± 14%

^a^ concentration [C content, mg/L and metal (μg/L)], mass fraction (%); ^b^ ([Me]/[DOC]) ratio, μmol-Me/g-OC.

**Table 4 ijerph-16-03398-t004:** Correlation coefficients of [Me]/[DOC] ratios with the selected optical indicators and C contents.

Samples	DOC	SUVA_254_	S_R_	FI
[Cd]/[DOC]_B__ulk_ (*n* = 24)	−0.67 ***	0.49 *	0.50 *	−0.22
[Ni]/[DOC]_Bulk_ (*n* = 24)	−0.68 ***	0.49 *	0.45 *	−0.21
[Cd]/[DOC]_LMHS_ (*n* = 24)	−0.85 ***	0.86 ***	0.61 ***	−0.51 **
[Ni]/[DOC]_LMHS_ (*n* = 24)	−0.48 *	0.68 ***	0.68 ***	−0.14
[Cd]/[DOC]_HMHS_ (*n* = 24)	−0.37	0.06	−0.15	0.22
[Ni]/[DOC]_HMHS_ (*n* = 24)	−0.18	−0.05	0.29	−0.16
[Cd]/[DOC]_Total_ (*n* = 72)	−0.248 *	0.233 *	0.021	0.114
[Ni]/[DOC]_Total_ (*n* = 72)	−0.361 **	0.346 **	0.568 ***	−0.263 *

* *p* < 0.05, ** *p* < 0.01, *** *p* < 0.001.

**Table 5 ijerph-16-03398-t005:** Predicted equations of [Me]/[DOC] ratio fitted with optical indicators.

Fitting Equations	R	*p*
[Cd]/[DOC] = −4.825 − 1.601 × S_R_ + 0.533 × SUVA_254_ + 3.159 × FI	0.509	<0.001
[Cd]/[DOC] = −3.838 − 1.767 × S_R_ + 0.515 × SUVA_254_ + 2.762 × FI − 0.0104 × [DOC]	0.535	<0.001
[Ni]/[DOC] = −10.936 +16.469 × S_R_ − 0.668 × SUVA_254_ + 6.635 × FI	0.652	<0.001
[Ni]/[DOC] = −5.497 + 15.554 × SR − 0.769 × SUVA254 + 4.476 × FI − 0.0573 × [DOC]	0.712	<0.001

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
