# Peer review of "Cadmium (Cd) and Nickel (Ni) Distribution on Size-Fractioned Soil Humic Substance (SHS)"

_ijerph, 2019, doi:10.3390/ijerph16183398_

Round 1

Reviewer 1 Report

This manu presents a study on the size fractioned soil humic substance, and its effect on soil cadmium and nickel distribution. The results showed that cadmium and nickel favor different fractions.

Basically the experimental design is reasonable, and the results presentation is accepatable.

My major concern is the language quality. Some comments can be found below, but I would suggest a professional language polishing.

Title: Probably delete 'heavy metals' and 'correlation with SHS optical indices'

Abstract

LIne 15, where--->whose?

Line 30, were both ....please clarify this sentence

Introduction

Line 50, in one study, but I notice that you cited six refs?

Line 53-53, if you want review these indices, pls expand it.

Line 59, I do not think it is necessary to use a new terminology here to describe the dissolved humic acid.

Overall, I feel the authors need to further clarify the logic behind introduction.

Results

Line 152, may say it like'pH, OM .. were,...., respectively'

Line 163, the total concentration of ...(Table 1)

Reviewer 2 Report

The manuscript presents results from a study of metal-binding capacity of soil humic acids. Although the study is interesting, it has problems with the experimental design and with writing, and it is too long for the amount of results presented. The abstract has too many abbreviations that don’t help reading, please avoid their use. In general, it is confuse and should be better structured. The introduction should be restructured, as it does not follow a logical sequence: the authors discuss humic substances properties, binding capacity of metals and methods for studying OM without a clear plan or order. Please reorganize following a clearer plan.

Methodology section is very confuse due to the inclusion of unnecessary information such as the description of the experimental design of a field experiment and lysimeters in lines 78-85. Also, methodology does not seem sound (please see below several of my comments).

Results are insufficiently reported, and in several parts the authors don’t present them at all, limiting themselves to report results from the literature.

Specific comments

Lines 19-20 Please remove or explain what these percentages represent

Lines 50-52 It is said “in one study…” but six references are given, please rewrite

Lines 83-84 Why was soils from 0-60 cm mixed? This makes no sense to me. If you want to extract humic substances, soil from the OM-rich surface should be used.

Line 84 Why three campaigns of sampling were undertaken? Or do you rather mean that three replicates were sampled?

Line 87 Why was soil sieved by 0.8 mm? Soil is routinely sieved by a 2-mm mesh before analysis

Line 93 Which soil was used for the extraction? It is said in line 85 that 24 soils were sampled.

Lines 110 and following “DOC” is a very precise term that refers to water-soluble OM, so I would suggests to replace simply by “C content” all throughout the text

Lines 157-162 Please rewrite, and differentiate clearly if you are reporting results from your own work or discussing results from other researchers

Lines 168-173 Same as previous comment.

Figures 1 and 2 Are these figures really necessary? Data could be added to Table 2.

Lines 215-223 Again, please differentiate clearly report of results from discussion.

Section 3.4 I would place this section before the section describing metals.

Reviewer 3 Report

The paper "Heavy Metals Cadmium (Cd) and Nickel (Ni) Distribution on Size-fractioned Soil Humic Substance (SHS) and Correlation with SHS Optical Indices" is interesting and well prepared. In my opinion, it needs minor revision.

The specific comments are as follows:

The title: In my opinion 'heavy metals' are unecessary as it is known thad Cd and Ni are heavy metals.

Abstract: L30 should be SHS DOC, please check.

BHS is a solution, so are the concentrations of BHS DOC and BHS metals expressed in mg per kg of soil or mg per kg of SHS. Please explain, because I am a little bit confused reading methodology and your results.

What was the reason of insufficient mass balance for other metals than Cd and Ni?. In my opinion, it would be also interesting to analyze results for Cu and Zn as they interact with humic compounds.

Could you explain more on calculation of Sr?

Could you explain more how did you obtain the predicted values of Me/DOC?

Round 2

Reviewer 1 Report

The authors have addressed most of my concerns, and the language quality improves largely for this submission. However I still feel that the research is very routine and not much new of the finding is there in the study. 

In the abstract, I would suggest the authors further improve the conclusion, as to me it only presents results.

Reviewer 2 Report

The authors have improved the structure of the manuscript and clarified some key questions raised by the reviewers. However, I still think that the manuscript is too long taking into account the amount of results presented, all from only one soil. The authors should work on the conclusions, that are still confuse, in particular in what concerns the "terrestrial source" of HS (I would suggest to remove those sentences from the conclusion).
